# Urinary actin, as a potential marker of sepsis-related acute kidney injury: A pilot study

**Dániel Ragán**[1,2]*, **Péter Kustán**[1], **Zoltán Horváth-Szalai**[1,3], **Balázs Szirmay**[1], **Beáta Bugyi**[4], **Andrea Ludány**[1], **Attila Miseta**[1], **Bálint Nagy**[2], **Diána Mühl**[2]

**1** Department of Laboratory Medicine, Medical School, University of Pécs, Pécs, Hungary, **2** Department of Anesthesiology and Intensive Therapy, Medical School, University of Pécs, Pécs, Hungary, **3** János Szentágothai Research Center, University of Pécs, Pécs, Hungary, **4** Department of Biophysics, Medical School, University of Pécs, Pécs, Hungary

* ragandaniel@hotmail.com

## Abstract

### Introduction

A major complication of sepsis is the development of acute kidney injury (AKI). Recently, it was shown that intracellular actin released from damaged tissues appears in the urine of patients with multiple organ dysfunction syndrome. Our aims were to measure urinary actin (u-actin) concentrations of septic and control patients and to test if u-actin levels could predict AKI and mortality.

### Methods

Blood and urine samples were collected from septic and sepsis-related AKI patients at three time points (T1-3): T1: within 24 hours after admission; T2: second day morning; T3: third day morning of follow-up. Patients with malignancies needing palliative care, end-stage renal disease or kidney transplantation were excluded. Serum and u-actin levels were determined by quantitative Western blot. Patients were categorized by the Sepsis-3 and KDIGO AKI classifications.

### Results

In our study, 17 septic, 43 sepsis-induced AKI and 24 control patients were enrolled. U-actin levels were higher in septic patients compared with controls during follow-up (p<0.001). At T1, the septic and sepsis-related AKI groups also showed differences (p<0.001), yet this increase was not statistically significant at T2 and T3. We also detected significantly elevated u-actin concentrations in AKI-2 and AKI-3 septic patients compared with AKI-1 septic patients (p<0.05) at T1 and T3, along with a significant increase in AKI-2 septic patients compared with AKI-1 septic patients at T2 (p<0.01). This tendency remained the same when referring u-actin to urine creatinine. Parameters of first-day septic patient samples could discriminate AKI from non-AKI state (AUC ROC, p<0.001): u-actin: 0.876; se-creatinine: 0.875. Derived cut-off value for u-actin was 2.63 μg/L (sensitivity: 86.0%, specificity: 82.4%).

**Data Availability Statement:** All relevant data are within the paper and its Supporting Information files.

**Funding:** The work was financially supported by the University of Pécs, Medical School, Hungary grant (KA-2018-17) and was also supported by the EFOP 3.6.1-16-2016-00004 project (Comprehensive Development for Implementing Smart Specialization Strategies) of the University of Pécs. Bálint Nagy was supported by the Thematic Excellence Program 2020—National Excellence Subprogram; Biomedical Engineering Project ("2020-4.1.1-TKP2020") of the University of Pécs. The funders had no role in study design, data collection and analysis, decision to publish, or preparation of the manuscript.

**Competing interests:** The authors have declared that no competing interests exist.

## Conclusion

U-actin may be a complementary diagnostic biomarker to se-creatinine in sepsis-related AKI while higher u-actin levels also seem to reflect the severity of AKI. Further investigations may elucidate the importance of u-actin release in sepsis-related AKI.

## Introduction

Effective treatment of sepsis is still a major challenge at the intensive care unit (ICU) with a 30% mortality rate [1]. Epidemiological research indicates an increasing incidence with a decreasing mortality in recent years [2, 3]. According to the latest Sepsis-3 definitions, sepsis is a life-threatening organ dysfunction caused by a dysregulated host response to infection [4].

Timely diagnosis and effective supportive therapy of sepsis is essential for a favorable outcome. Serum procalcitonin (PCT) and high-sensitivity C-reactive protein (hs-CRP) are still the gold standard inflammatory markers in the current sepsis guidelines [4, 5]. Besides hs-CRP and PCT, more than 200 potential (mostly serum) sepsis biomarkers have been investigated in recent years but none of them showed sufficient performance for routine clinical practice [6, 7].

Acute kidney injury (AKI) is one of the most frequent organ dysfunctions associated with sepsis or septic shock with a mortality rate up to 60% [8]. AKI refers to an abrupt (mostly reversible) decrease in kidney function resulting in the retention of numerous waste products and the dysregulation of extracellular volume. The Kidney Disease Improving Global Outcomes (KDIGO) classification is widely used for the diagnosis of AKI based on the elevation of serum creatinine levels and the decrease in urine output [9]. However, the changes in these parameters do not reflect the extent of kidney damage, therefore more sensitive and specific laboratory biomarkers are needed for the early diagnosis and prediction of AKI. Regarding this, serum neutrophil gelatinase-associated lipocalin (NGAL), kidney injury molecule-1 (KIM-1) and cystatin-C are extensively investigated novel biomarkers [10, 11]. Recent studies and reviews indicate other novel kidney damage biomarkers investigated in other clinical conditions (e.g. during cardiac surgery or ischemia-reperfusion injury), such as liver-type fatty acid binding protein (L-FABP), interleukin 18 (IL-18), hemojuvelin (HJV) and osteopontin (OPN), while other so-called "pre-injury phase" stress biomarkers (e.g. tissue inhibitor of metalloproteinases-2 (TIMP-2) and insulin-like growth factor-binding protein 7 (IGFBP7)) showed promising results in the beginning stages of moderate AKI [12–14]. However, there is still a lack of knowledge on urinary biomarkers, including a ubiquitous intracellular protein, called actin.

Actin is a globular protein (MW = 42 kDa) existing in monomeric/globular (G-actin) and in polymeric/filamentous (F-actin) forms. In acute tissue injuries the released extracellular actin is found to be highly toxic in the circulation due to its spontaneous polymerizing tendency causing unfavorable effects on the coagulation system. Gelsolin and Gc-globulin are the most important proteins of the so-called actin scavenger system which is responsible for binding and depolymerizing extracellular actin in the circulation, thus making the urinary appearance of these protein complexes unlikely [15–18]. However, an earlier study indicates that actin could be detected in the urine of kidney transplant patients with sustained AKI [19].

Therefore, we aimed to be the first to describe the time course of urinary actin (u-actin) in sepsis and sepsis-related AKI by ICU patients. We hypothesized that elevated u-actin concentrations might be useful regarding the early diagnosis of sepsis-related AKI as well as a

predictive marker of the outcome. Unfortunately, there are no commercial diagnostic kits available yet for serum and/or urinary actin quantification. Therefore, first we had to develop a sensitive and specific Western blot method for the quantitative analysis of actin levels.

## Materials and methods

### Study design

Patients with acutely confirmed sepsis or sepsis-related AKI were enrolled consecutively in our follow-up study conducted between January 2016 and December 2019 at the Department of Anesthesiology and Intensive Therapy (Medical School, University of Pécs, Hungary). All patients or their next-of-kin were given detailed information regarding our study protocol while written consent was also obtained from all. Exclusion criteria were patients with malignancies needing palliative care, end-stage renal disease with chronic dialysis or kidney transplantation, under 18 years of age or unobtainable consent. The study protocol was approved by the Regional Research Ethics Committee of the University of Pécs (no. 4327.316-2900/ KK15/2011) in accordance with the 7[th] revision of the Helsinki Declarations (2013).

The diagnosis of sepsis and AKI were established using the Sepsis-3 definitions and KDIGO guidelines, respectively [4, 5, 9]. Inclusion criteria for sepsis were signs of organ dysfunction shown in increased Sequential Organ Failure Assessment (SOFA) score ($>2$; median: 10), elevated serum PCT levels ($>2$ ng/mL; median: 9.865 ng/mL) and a suspected or microbiologically confirmed infection. Patients with elevated serum creatinine levels ($\geq 1.5$—fold increase from the baseline in the last 7 days or $\geq 26.5$ μmol/L increase within 48 hours) or with decreased urine output ($<0.5$ mL/kg/h for 6 hours) were regarded to have AKI. Management of sepsis and sepsis-related AKI followed the international guidelines of the 2016 Surviving Sepsis Campaign regarding vasopressor, respiratory, anticoagulation and hydrocortisone therapy, along with feeding, ulcer prophylaxis and sedation [5]. All patients received adequate fluid resuscitation and ex juvantibus broad spectrum antimicrobial therapy guided by the clinical presentation for 24 to 72 hours, which was later modified based on the results of microbiological investigations. Defined end points were death or withdrawal of consent during the sample collection period.

Regarding disease severity and mortality, the first-day values of Simplified Acute Physiology Score II (SAPS II), Acute Physiology and Chronic Health Evaluation II (APACHE II) and SOFA scores were determined. Using 30-day mortality data, patients were divided into survivor and non-survivor groups.

Outpatients from the Department of Ophthalmology (Medical School, University of Pécs, Hungary) served as controls. Exclusion criteria for control patients were lack of consent, acute inflammation (hs-CRP $>5$ mg/L), infectious disease or kidney disease.

### Sampling

Blood and urine samples were taken from septic patients at the ICU at three time points (T1-3): T1: within 24 hours after admission; T2: second day morning of follow-up; T3: third day morning of follow-up. Venous blood (5 mL) was taken from every septic patient from central venous catheter into plastic blood collection tubes with accelerator gel (BD Vacutainer, Franklin Lakes, NJ, USA), while urine (6 mL) was obtained simultaneously from the bladder catheter using plain plastic tubes (Sarstedt AG, Nümbrecht, Germany). Not more than one sample (venous blood, midstream spot urine) was collected from control patients. Urine and clotted blood samples were centrifuged (10 min, 1500 g) and supernatants were treated with electrophoresis sample buffer [20] then heated at 100˚C for 5 minutes. Serum and urinary actin were

determined by quantitative Western blot (see below). Native and denatured sample aliquots were stored at -70˚C until analysis.

## Laboratory analysis

Serum parameters including total protein (se-TP), albumin, creatine kinase (se-CK), kidney function markers (se-creatinine), platelet count (PLT), inflammatory parameters (white blood cell count (WBC), hs-CRP, PCT) and basic urinary markers (total protein (u-TP), u-albumin, urine creatinine (u-creatinine)) were determined using automated routine procedures at our accredited laboratory (Department of Laboratory Medicine, Medical School, University of Pécs, Hungary). Urinary orosomucoid (u-ORM) and urinary cystatin-C (u-CysC) were also measured using automated immune turbidimetric assays (Cobas 8000/c502 module (Roche Diagnostics GmbH, Mannheim, Germany)) worked out and validated in our laboratory [21, 22].

## Determination of urinary actin

Serum and urine samples were separated with a 10% SDS-polyacrylamide gel electrophoresis according to Laemmli [20]. Serum actin (se-actin) levels were determined by a quantitative enhanced chemiluminescence (ECL) Western blot based on the work of Lee et al. and Horváth-Szalai et al. [23–25]. This method was optimized and adapted for the determination of u-actin concentrations. A polyclonal pan-anti-actin primary antibody (1:1000 dilution; Rabbit Anti-Human Actin, N-terminal, ref.no: A2103, Sigma-Aldrich, St. Louis, MO, USA) and a horseradish peroxidase (HRP)-labeled secondary antibody (1:4000 dilution; Goat Anti-Rabbit IgG, Cat. #31460, Thermo Scientific, Rockford, IL, USA) were used. The calibration of Western blot was done by running a dilution series of a purified rabbit skeletal muscle extract of known G-actin concentration (Department of Biophysics, Medical School, University of Pécs, Hungary). A 3-part dilution series of a serum sample from a healthy individual was applied in every gel as an internal standard which had been quantified by the mentioned rabbit skeletal muscle extract calibrator. Therefore, actin concentrations could be calculated in each serum and urine sample based on the light signals being directly proportional to the internal standard series applied in every gel.

## Statistical analysis

The IBM SPSS Statistics for Windows, Version 22 (IBM Corp., NY, USA) software was used for statistical analysis. Non-parametric tests were used since our data did not show normal distribution by the Kolmogorov-Smirnov and Shapiro-Wilk tests. The control and septic patient groups were compared using Chi-square test for qualitative data and Mann-Whitney U-test for quantitative data. Friedman's ANOVA with post hoc Dunn tests were carried out to compare the data of different time points in every patient group. Diagnostic and prognostic values of serum and urinary parameters were evaluated by receiver operating characteristic (ROC) curves. Relationships between quantitative variables were analyzed by Spearman's rank correlation test. Qualitative data are shown as frequencies and percentages (%) while quantitative data are shown as medians and interquartile ranges (IQR). Statistical significance was set at $p < 0.05$. P values were adjusted according to Bonferroni during the analyses of multiple comparisons.

# Results

## Patients' demographic and laboratory data

In the present study, 24 control individuals, 17 septic patients and 43 patients with sepsis-related AKI were enrolled. Basic demographic and admission laboratory data are shown in

**Table 1. Patients' demographic and admission laboratory data.**

| | Control (n = 24) | Sepsis (n = 17) | Sepsis + AKI (n = 43) | p value |
|---|---|---|---|---|
| Age (years) | 54 (48–57) | 73 (64–78) | 64 (53–70) | <0.05[a,b,c] |
| Males, n (%) | 10 (41.7) | 8 (47.1) | 33 (76.7) | <0.05[b,c] |
| **Major comorbidities, n (%)** | | | | |
| Cardiovascular disease | 11 (45.8) | 15 (88.2) | 30 (69.8) | <0.05[a] |
| Type-2 diabetes mellitus | 4 (16.7) | 3 (17.6) | 15 (34.9) | n.s. |
| Chronic kidney disease | 0 | 3 (17.6) | 5 (11.6) | <0.05[a] |
| KDIGO-CKD stages, n (GFR/Albuminuria) | - | 3 (G3b/A1) | 1 (G2/A1); 1 (G3a/A2); 3 (G4/A2) | - |
| Pulmonary disease | 1 (4.2) | 3 (17.6) | 11 (25.6) | <0.05[b] |
| Immunological disease | 2 (8.3) | 1 (5.9) | 3 (6.9) | n.s. |
| Malignancy | 0 | 4 (23.5) | 12 (27.9) | <0.05[a,b] |
| **Admission laboratory data** | | | | |
| se-TP (g/L) | 71.6 (67.4–73.7) | 40.9 (38.6–46.9) | 46.9 (42.2–50.4) | <0.05[a,b] |
| se-albumin (g/L) | 42.7 (40.1–46.3) | 21.2 (19.5–25.9) | 23.1 (19.2–26.6) | <0.05[a,b] |
| se-creatinine (μmol/L) | 71 (67–76) | 91 (67–121) | 182 (142–328) | <0.05[a,b,c] |
| se-CK (U/L) | 40 (54–73) | 198 (145–649) | 305 (88–1047) | <0.05[a,b] |
| WBC (G/L) | 5.9 (4.7–6.9) | 13.2 (9.8–17.5) | 16.4 (9.9–20.5) | <0.05[a,b] |
| PLT (G/L) | 265 (226–301) | 223 (159–265) | 162 (95–301) | <0.05[a,b] |
| hs-CRP (mg/L) | 1.5 (0.6–1.8) | 304.2 (188.9–372.4) | 276.1 (138.6–388.7) | <0.05[a,b] |
| PCT (ng/mL) | - | 7.9 (3.1–20.6) | 23.6 (6.1–59.9) | <0.05[c] |
| se-actin (mg/L) | 0.75 (0.6–0.9) | 0.68 (0.4–1.1) | 0.84 (0.5–1.2) | n.s. |
| u-TP (mg/L) | 60 (40–90) | 230 (50–425) | 440 (240–1020) | <0.05[a,b,c] |
| u-albumin (mg/L) | 3.5 (2.0–12.5) | 17.1 (3.4–54.4) | 61.8 (26.8–169.2) | <0.05[a,b,c] |
| u-ORM (mg/L) | 1.2 (0.3–3.2) | 32.6 (17.8–87.5) | 72.1 (23.7–144.8) | <0.05[a,b] |
| u-CysC (mg/L) | 0.06 (0.03–0.08) | 0.46 (0.1–1.1) | 1.04 (0.2–3.5) | <0.05[a,b] |
| u-actin (μg/L) | 0.78 (0.3–1.9) | 1.27 (0.6–2.5) | 9.5 (3.2–23.7) | <0.05[b,c] |
| u-actin/u-creatinine (μg/mmol) | 0.13 (0.1–0.2) | 0.6 (0.4–1.4) | 2.59 (0.6–10.4) | <0.05[a,b,c] |
| u-actin/u-TP (%) | 0.17 (0.1–0.3) | 0.08 (0.04–0.26) | 0.21 (0.1–1.1) | <0.05[c] |

Continuous variables are shown as median (25th– 75th percentiles) and categorical variables are expressed as a number (percentage). Mann-Whitney U and Chi-square tests were used for data comparison between patient groups. Level of significance is set at p<0.05; n.s.: non-significant. Superscript lowercase letters refer to post-hoc analyses:

a: p<0.05 between Control and Sepsis

b: p<0.05 between Control and Sepsis + AKI

c: p<0.05 between Sepsis and Sepsis + AKI groups.

Table 1. A moderate difference (p<0.05) was found between the patient groups regarding age, gender and the majority of comorbidities. A significant difference (p<0.05) was found between the control and septic patient groups in se-TP, se-albumin, se-CK, WBC, PLT, hs-CRP, u-CysC and u-ORM levels. Admission values of se-creatinine, u-TP, u-albumin, and u-actin/u-creatinine were significantly higher (p<0.05) in the septic and sepsis-related AKI groups compared with the control patients. Se-PCT (p<0.05), u-actin (p<0.05) and u-actin/u-TP (p<0.05) levels were also significantly higher between the septic and sepsis-related AKI patients.

## Septic patients' clinical data

Major differences in therapeutic requirements of 60 septic patients are the following: 38 needed mechanical ventilation, 22 needed oxygen supplementation (median Horowitz

**Table 2. Clinical and microbiological data of septic patients.**

|  | Sepsis (n = 17) | Sepsis + AKI (n = 43) | p value |
|---|---|---|---|
| Age (years) | 73 (64–78) | 64 (53–70) | <0.05 |
| Males, n (%) | 8 (47.1) | 33 (76.7) | <0.05 |
| **Cause of admission** | | | |
| Internal medicine origin, n (%) | 3 (17.6) | 22 (51.1) | <0.05 |
| Surgical origin, n (%) | 14 (82.4) | 21 (48.9) | <0.05 |
| ICU treatment days | 9 (5–17) | 6 (3–15) | n.s. |
| 30-day mortality, death (%) | 8 (47.1) | 28 (65.1) | n.s. |
| AKI requiring RRT, n (%) | - | 18 (41.9) | - |
| **Clinical prognostic scores** | | | |
| APACHE II score | 15 (11–17) | 24 (17–27) | <0.05 |
| SAPS II score | 41 (33–46) | 55 (41–64) | <0.05 |
| SOFA score | 8 (7–11) | 11 (9–13) | <0.05 |
| **Organ dysfunctions, n (%)** | | | |
| 1 | 4 (23.5) | 5 (11.6) | n.s. |
| 2 | 4 (23.5) | 7 (16.3) | n.s. |
| ≥3 | 9 (53.0) | 31 (72.1) | n.s. |
| **Identified pathogens, n (%)** | | | |
| Unidentified | 6 (35.3) | 11 (25.6) | n.s. |
| Gram-positive bacteria | 1 (5.9) | 6 (13.9) | n.s. |
| Gram-negative bacteria | 4 (23.5) | 10 (23.3) | n.s. |
| Fungi | 1 (5.9) | 4 (9.3) | n.s. |
| Mixed | 5 (29.4) | 12 (27.9) | n.s. |

Continuous variables are shown as median (25th– 75th percentiles) and categorical variables are expressed as a number (percentage). Mann-Whitney U and Chi-square tests were performed for data comparison between patient groups. Level of significance is set at p<0.05; n.s.: non-significant. Abbreviations: RRT: renal replacement therapy.

quotient: 221.2), 51 needed vasopressor support, 12 had liver failure, 13 had thrombocytopenia, 30 had elevated lactate levels (>2 mmol/L) and 47 needed hydrocortisone supplementation. Ventilated patients received Propofol or Dexmedetomidine sedation during the early stage of severe respiratory failure. Regarding differences between the septic and sepsis-related AKI groups, more patients needed ICU hospitalization due to surgical interventions (e.g. esophageal or pancreatic cancer surgery, peritonitis) than internal medicine complications (e.g. pneumonia) (p<0.05). No significant difference was found between the septic patient groups regarding the length of ICU stay and 30-day mortality, however, 30-day mortality was higher in the AKI-1 (8 patients, 80.0%) and AKI-2 (10 patients, 71.4%) stage septic groups compared with the AKI-3 (10 patients, 52.6%) stage septic and sepsis (8 patients, 47.1%) groups. Moreover, 41.9% of sepsis-related AKI patients required some kind of renal replacement therapy (RRT). Regarding the clinical prediction scores, sepsis-related AKI patients had considerably higher values (APACHE II: p<0.05; SAPS II: p<0.05; SOFA: p<0.05) compared with the sepsis group while multiple organ dysfunction syndrome (MODS) was also a more common complication in the former group (69.8%) than in the latter (53.0%). The most frequent organ dysfunctions–besides AKI–in the sepsis-related AKI and sepsis patient groups were sepsis-induced hypotension (78.1% vs. 76.5%), followed by acute lung injury (48.7% vs. 52.9%), thrombocytopenia (25.6% vs. 11.8%) and acute liver failure (25.6% vs. 5.9%). Additional clinical data of septic patients are presented in Table 2 and in S1 Table as well.

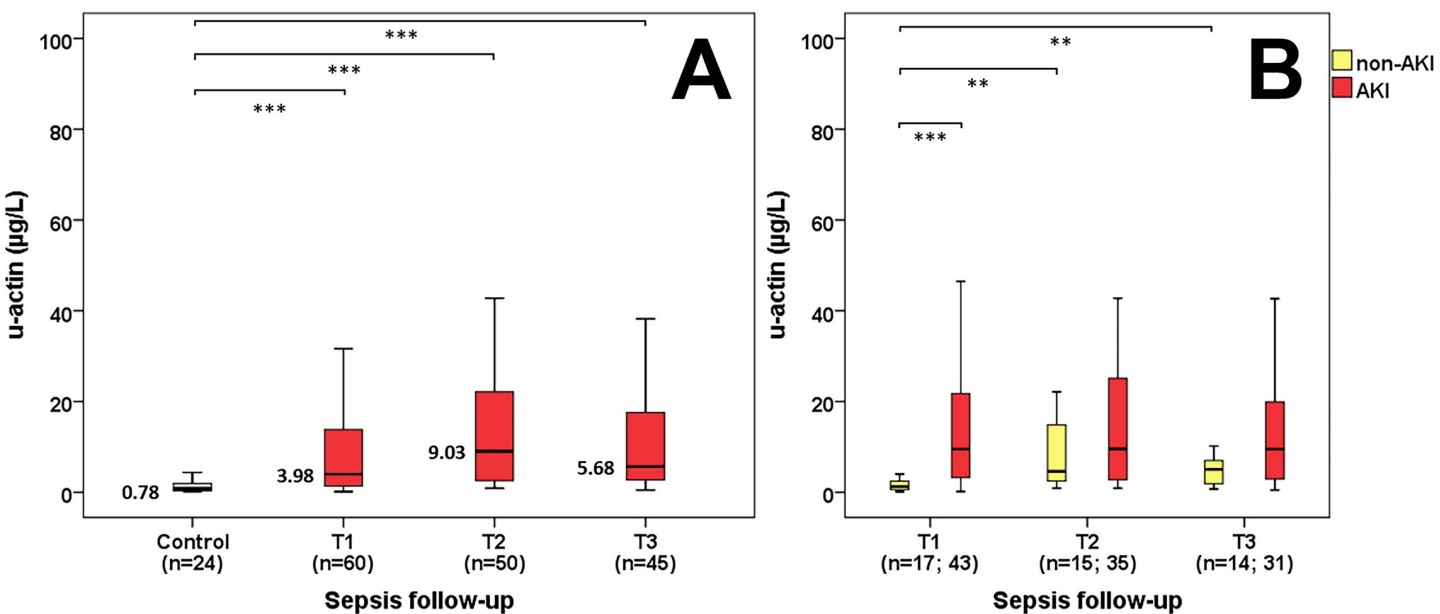

**Fig 1. Urinary actin in sepsis.** U-actin levels of control and septic patients (A) along with sepsis and sepsis-related AKI patients (B) during follow-up. n: sample count. **p<0.01; ***p<0.001.

## Serum and urinary actin levels in control, septic and sepsis-related AKI patients

No considerable difference was observed in serum actin levels between the control and septic patient groups by comparing the first (T1 median: 0.75 vs. 0.8 mg/L, p = 0.757), second (T2 median: 0.75 vs. 0.78 mg/L, p = 0.584) and third (T3 median: 0.75 vs. 0.79 mg/L, p = 0.608) day serum samples during follow-up. However, a significant increase in u-actin levels was discovered between the control and septic patients during the first (T1 median: 0.78 vs. 3.98 µg/L, p<0.001), second (T2 median: 0.78 vs. 9.03 µg/L, p<0.001) and third (T3 median: 0.78 vs. 5.68 µg/L, p<0.001) day of follow-up (Fig 1A). This tendency was also explicit between the septic and sepsis-related AKI groups at the first day (T1 median: 1.27 vs. 9.52 µg/L, p<0.001), but this difference was not statistically significant by the second (T2 median: 4.61 vs. 9.55 µg/L, p = 0.368) and third (T3 median: 5.05 vs. 9.51 µg/L, p = 0.220) day of follow-up (Fig 1B). Regarding the AKI stages, sepsis-related AKI patients were categorized retrospectively based on the KDIGO guidelines. We also detected significantly elevated u-actin concentrations on the first day regarding AKI-1 and AKI-2 (T1 median: 3.16 vs. 10.78 µg/L, p<0.01) along with AKI-1 and AKI-3 (T1 median: 3.16 vs. 11.55 µg/L, p<0.01) septic patients, while these changes were also statistically significant on the second day regarding AKI-1 and AKI-2 (T2 median: 3.22 vs. 9.68 µg/L, p<0.01) septic patients and on the third day regarding AKI-1 and AKI-2 (T3 median: 3.66 vs. 13.76 µg/L, p<0.05) along with AKI-1 and AKI-3 (T3 median: 3.66 vs. 12.92 µg/L, p<0.05) septic patients (Fig 2A). This tendency remained the same when referring u-actin values to u-creatinine on the first day regarding AKI-1 and AKI-2 (T1 median: 0.47 vs. 3.74 µg/mmol, p<0.05) along with AKI-1 and AKI-3 (T1 median: 0.47 vs. 6.16 µg/mmol, p<0.05) septic patients, on the second day regarding AKI-1 and AKI-2 (T2 median: 0.87 vs. 4.53 µg/mmol, p<0.05) septic patients and on the third day regarding AKI-1 and AKI-3 (T3 median: 0.84 vs. 5.37 µg/mmol, p<0.05) septic patients (Fig 2B).

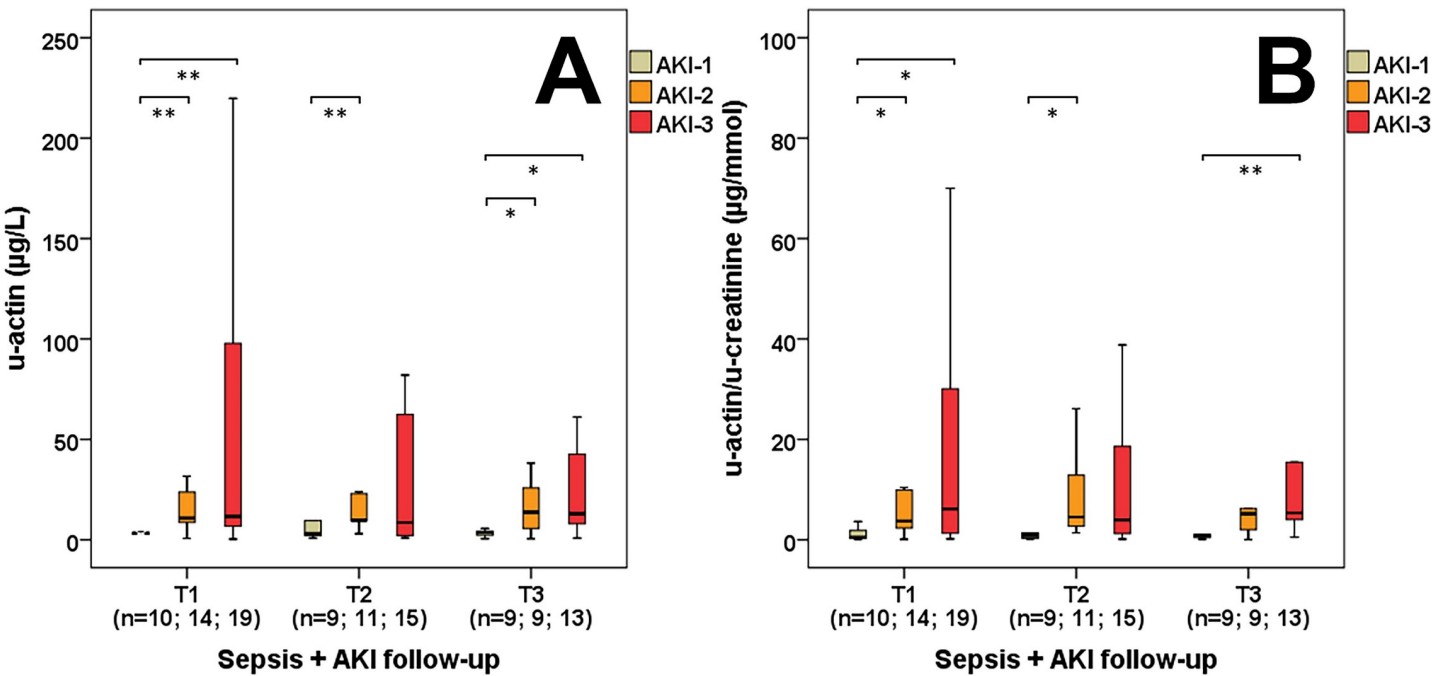

**Fig 2. Urinary actin in sepsis-related AKI.** U-actin (A) and u-actin/u-creatinine (B) levels of the individual sepsis-related AKI stages during follow-up. n: sample count. *p<0.05; **p<0.01.

### Survival data and distinctive power of urinary actin in sepsis

No significant difference was found in u-actin levels between survivors and non-survivors based on 30-day mortality data during follow-up (Fig 3A). The diagnostic performance of

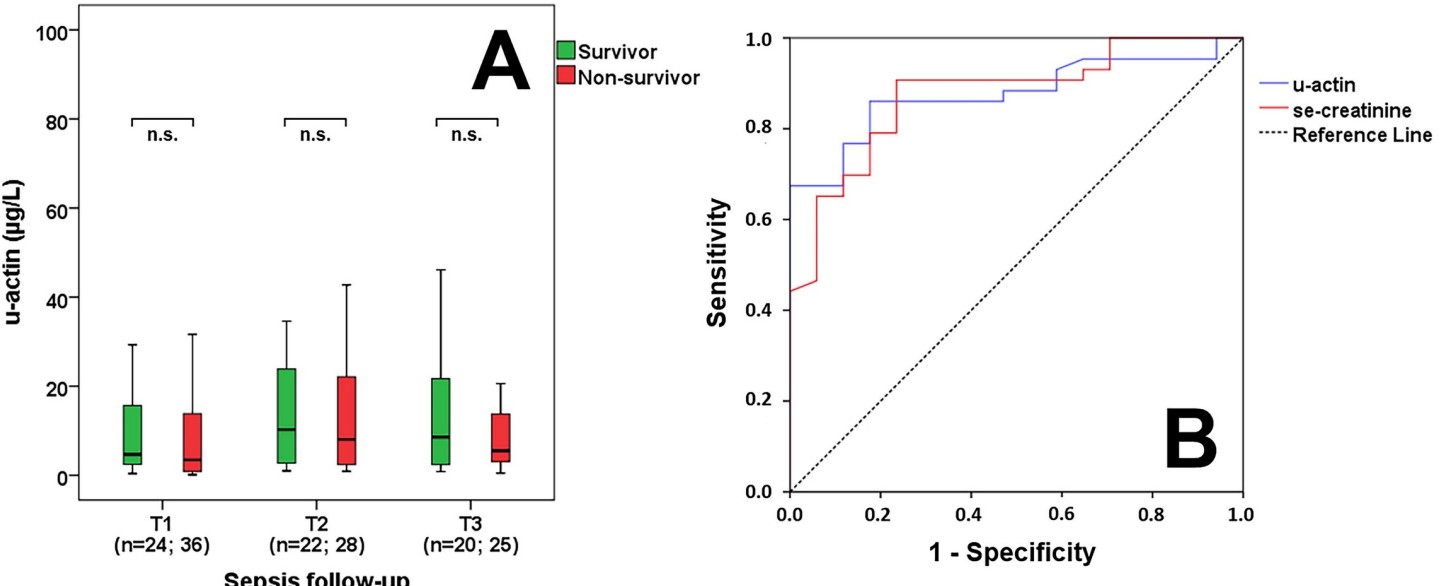

**Fig 3. Survival and predictive power of urinary actin.** U-actin levels in survivor and non-survivor septic patients based on 30-day mortality during follow-up (A). Receiver operating characteristic (ROC) curves of admission laboratory parameters for distinguishing sepsis from sepsis-related AKI (B). n: sample count. n.s.: non-significant.

first-day u-actin levels in sepsis-related AKI was assessed using ROC analysis (Fig 3B). For discerning all sepsis-related AKI from septic patients without AKI, area under the curve (AUC) values were found to be the following: u-actin: 0.876 (p<0.001); se-creatinine: 0.875 (p<0.001); Derived cut-off values were: u-actin 2.63 µg/L (sensitivity: 86.0%, specificity: 82.4%); se-creatinine: 111 µmol/L (sensitivity: 90.7%, specificity: 76.5%).

## Correlations

In our study, data from all sample collection time points were used for calculating correlations, while the absolute u-actin levels were presented in most cases due to their strong correlation to u-actin/u-creatinine ($\rho$ = 0.898, p<0.001). U-actin levels showed moderate correlation (p<0.001) with u-TP ($\rho$ = 0.489), u-albumin ($\rho$ = 0.617), se-creatinine ($\rho$ = 0.371), u-CysC ($\rho$ = 0.434) and u-ORM ($\rho$ = 0.367). Weak correlations (p<0.01) were observed between u-actin and se-actin ($\rho$ = 0.272) and WBC ($\rho$ = 0.223), respectively. No further associations were found with other inflammatory or clinical parameters.

## Discussion

The early diagnosis and effective therapy of sepsis and sepsis-related AKI are essential for a successful recovery. However, the currently used biomarkers (sepsis: PCT, hs-CRP; AKI: se-creatinine, urine output) show several limitations regarding the diagnosis and prognosis of sepsis and AKI, hence a multi-marker approach involving novel laboratory markers may prove to be beneficial in achieving a favorable outcome [6, 7, 10, 11].

There is a growing body of evidence indicating the importance of decreased actin scavenger protein concentrations and/or elevated se-actin levels in various clinical conditions (e.g. trauma, acute liver failure, myocardial infarction, sepsis) [26–29]. Our previous study showed that the increase of se-actin was inversely proportional to the amounts of actin scavenger proteins in the circulation which was associated with increased mortality rate [25].

Since urine is noninvasively available, it is becoming a popular source for disease biomarker studies. Several studies have already examined actin in the circulation, however, the importance of u-actin has not yet been extensively investigated [17, 24, 25, 30]. Only one study carried out by Kwon et al. investigated u-actin as a potential AKI marker. They found increased u-actin/u-creatinine ratios in patients after cadaveric kidney transplantation (u-actin/u-creatinine levels were significantly higher in patients with sustained AKI compared to patients recovering from AKI). However, the origin of this considerable u-actin release has not been clarified, the ischemic injury of renal allografts was assumed to be the main cause of AKI [19].

One of the main focuses of our study was to develop a sensitive method for detecting actin in the urine samples of septic patients. In accordance with previous findings, se-actin levels were slightly elevated in our septic patients compared with controls [25]. In contrast to the study conducted by Kwon et al., we could also detect low concentrations of u-actin in each control sample as well. This might be due to our improved actin-detecting technique. Significantly higher u-actin levels were found in the urine samples of septic patients compared with the control group, yet this increase was also explicit between the sepsis and sepsis-related AKI groups, especially in sepsis-related AKI patients needing renal replacement therapy (RRT).

During the 3-day follow-up period, no significant difference was found in u-actin levels between survivors and non-survivors. These findings suggest that u-actin may not be a suitable marker for sepsis or sepsis-related AKI mortality prediction. Moreover, no correlation was observed between u-actin and the conventional inflammatory markers (hs-CRP, PCT) or clinical scores (APACHE II, SAPS II, SOFA). Therefore, our results contradict the data available

from large multi-center clinical trials indicating better performance of currently used sepsis markers regarding mortality prediction [4, 5].

Our study has several limitations. As far as we know, before our study nobody investigated u-actin in sepsis or in sepsis-related AKI, thus we aimed to be the first to explore this promising and interesting field by conducting a small sample size pilot study to establish some baseline data. Since our study was performed as single center study (16 bedded central ICU), we had very limited capacities for consecutive patient recruitment, hence the patient groups were different regarding age and gender, while the study took a longer time period to achieve this relatively small sample size. Differences in sample size between control, septic and sepsis-related AKI groups might reduce the power of comparison, although non-parametric tests (e.g. Mann-Whitney U test) could be used well with unequal sample sizes. There was a slightly variable time interval (mostly within 12 hours) in the actual first day sample collection, as most patients were admitted at night or in the late afternoon before taking the first sample on the next morning.

In our research, the absolute u-actin levels are shown in most of our results due to their strong correlation to u-actin/u-creatinine. We are aware of the practice that novel urinary AKI biomarkers have to be normalized to u-creatinine concentrations in order to control variations in urine flow rate. However, there is a growing body of evidence challenging the importance of u-creatinine normalization in AKI since the excretion rates of u-creatinine and other novel urinary AKI biomarkers may be affected differently during an acute severe condition [31–34].

A moderate correlation with u-TP, u-albumin and se-creatinine suggest that elevated u-actin levels could be the consequence of severe glomerular injury potentially caused by sepsis-related systemic cellular damage with an excessive release of free extracellular actin. However, the increase of u-actin might not be the direct result of excessive free actin release due to a weaker correlation with se-actin and a moderate correlation with u-CysC indicating a tubular dysfunction as well during the development of AKI. Se-CK did not show significant correlations with se-actin or u-actin suggesting the cause of increased u-actin is not extensive muscle damage. Since renal ischemia is common during sepsis, it could also contribute to the development of sepsis-related AKI, as postulated by Kwon et al. [19].

As actin is bound to gelsolin or Gc-globulin in the circulation, it is unlikely to appear in the urine. However, u-actin could appear in the urine due to both severe glomerular or tubular injury, so it seems that the elevation of u-actin assumes severe kidney injury. As KIM-1 or NGAL are known examples of damage biomarkers in AKI according to recent reviews, u-actin may also be classified into this group [12, 13]. We suspect that u-actin might provide more accurate information regarding the extent of kidney injury compared with serum creatinine. Therefore, a multi-marker approach including u-actin and the measurement of various damage biomarkers (e.g. urine Cystatin C, KIM-1, NGAL, IL-18) may provide valuable information regarding the more accurate staging of sepsis-related AKI.

Since different forms of RRT–especially continuous renal replacement therapy (CRRT)–represent a better modality for the management of severe sepsis-related AKI, there is no clear evidence demonstrating the adequate time point of initiation of RRT in sepsis-related AKI [13, 35–37]. Since actin is either bound to the actin scavenger proteins in the circulation or immediately starts forming filaments during severe tissue injury, it could not be removed from the circulation by RRT, yet the increase of u-actin could yield valuable information regarding the extent of kidney damage and the need for the early initiation of RRT in sepsis-related AKI.

In the future, we should extend the sampling period to 5–7 days while increasing the number of critically ill patients, especially patients needing RRT with forming subgroups due to the heterogenity of sepsis in order to draw more accurate conclusions. Exploring other patient groups with different kinds of kidney disease (e.g. CKD, glomerulonephritis) might provide

additional information regarding the appearance of actin in the urine. As this is the first study investigating the importance of u-actin in sepsis-related AKI, there are no commercially available diagnostic kits for serum and/or urinary actin quantification. For this reason, we developed a highly sensitive Western blot method evaluated by an ECL technique. Since Western blot is a quite expensive and time-consuming laboratory technique, its routine clinical utility is questionable, hence the development of a more rapid and efficient laboratory method (e.g. ELISA, POCT) is necessary.

## Conclusions

U-actin may serve as a complementary diagnostic biomarker to se-creatinine in sepsis-related AKI while higher u-actin levels also seem to reflect the severitiy of AKI. As the optimal time point for the initiation of RRT in sepsis-related AKI is still controversial, u-actin might provide valuable information regarding this issue. Further investigations may elucidate the importance of increased u-actin release in sepsis-related AKI.

## Supporting information

**S1 Table. The anonymous dataset of 84 patients.** Additional demographic, clinical and laboratory data of control, septic and sepsis-related AKI patients.
(XLSX)

## Acknowledgments

A preliminary version of the study has been presented as part of a mini-review [38] and as an abstract at the 59th National Congress of the Hungarian Society of Laboratory Medicine, 2018, Pécs, Hungary [39]. We express our special thanks for the invaluable help of the nurses and the colleagues from every participating institute.

## Author Contributions

**Conceptualization:** Péter Kustán.

**Data curation:** Dániel Ragán, Zoltán Horváth-Szalai.

**Formal analysis:** Dániel Ragán.

**Funding acquisition:** Attila Miseta.

**Investigation:** Péter Kustán, Balázs Szirmay.

**Methodology:** Dániel Ragán, Zoltán Horváth-Szalai, Balázs Szirmay, Beáta Bugyi.

**Resources:** Beáta Bugyi, Attila Miseta.

**Supervision:** Andrea Ludány, Attila Miseta, Bálint Nagy, Diána Mühl.

**Validation:** Diána Mühl.

**Writing – original draft:** Dániel Ragán.

**Writing – review & editing:** Andrea Ludány, Bálint Nagy, Diána Mühl.

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
