## [Decision Letter · Decision Letter 0]

24 Jun 2021

PONE-D-21-17545

Urinary actin, as a potential marker of sepsis-related acute kidney injury: a pilot study

PLOS ONE

Dear Dr. Ragan,

Thank you for submitting your manuscript to PLOS ONE. After careful consideration, we feel that it has merit but does not fully meet PLOS ONE’s publication criteria as it currently stands. Therefore, we invite you to submit a revised version of the manuscript that addresses the points raised during the review process.

ACADEMIC EDITOR:  Thank you for this manuscrpit. The topic is interesting, but there many concerns. 

In my personal opinion it is not clear how  the patients were included. There are no informations about patient therapy. You should discuss the timing of this marker in identyfing the AKI. 

No conflicts between the reviews.

See comments below.

We look forward to receiving your revised manuscript.

Kind regards,

Martina Crivellari

Academic Editor

PLOS ONE

Journal Requirements:

Additional Editor Comments (if provided):

Reviewers' comments:

Reviewer's Responses to Questions

**Comments to the Author**

1. Is the manuscript technically sound, and do the data support the conclusions?

Reviewer #1: Yes

Reviewer #2: Yes

2. Has the statistical analysis been performed appropriately and rigorously? 

Reviewer #1: Yes

Reviewer #2: Yes

3. Have the authors made all data underlying the findings in their manuscript fully available?

Reviewer #1: Yes

Reviewer #2: Yes

4. Is the manuscript presented in an intelligible fashion and written in standard English?

Reviewer #1: Yes

Reviewer #2: Yes

5. Review Comments to the Author

Reviewer #1: Dear author, I have read your study with great enthusiasm, as evaluating an early diagnosis of AKI in septic patients can improve the care provided to these patients, however I have some considerations about the study that need to be clarified.

By parts, I request majors comments:

Title: why include a pilot study in the title? Nowhere in the text was that explained.

Abstract: Instead of describing the sample size in the methods why didn't you describe which patients were included and excluded from the study? The sample size must be considered in the results.

Introduction: The first paragraph separately presents the subject of sepsis and your manuscript is related to sepsis-AKI, so I suggest shortening the information in this paragraph, because it is out of context.

In the introduction there is a lack of information about AKI biomarkers, it is subtly described.

Methods: In this item we have the biggest weakness of the study. First of all, how were the patients included? consecutively? The study period is very long and the number of patients seems to be small, why did this happen? Was there a random choice of patients to participate in the study or as admitted were they involved in the study? This information is relevant because it can be a bias in patient selection.

Inclusion and exclusion criteria are poorly described. Could patients with previous renal failure or dialysis participate in the study? Could patients in palliative care without treatment perspective also participate in the study? Could you better explain all these criteria and references for the decision on inclusion and exclusion?

This issue impacts the final results found in the study.

Second point that is not clear, the sample size included in the study, why did this distribution occurre 17 sepsis, 43 sepsis-AKI and 24 control? How were these numbers found? Are these numbers enough to achieve the results found?

Third, some relevant information needs to be collected in the study, for example fluid balance, use of vasopressor drugs and the need for mechanical ventilation. These issues could directly impact the outcomes, it is important to know if patients were treated equally and what the main protocols for these events would be.

Finally, why were these sample collection periods adopted for the realization of biomarkers? How were the correlations between biomarkers analyzed based on these different periods of material collection?

Statistical analysis

What was the outcome parameter used in ROC and how was the reference point of the markers reached?Could you describe this calculation better?

In the analysis of multiple comparisons, the value of P was not corrected? Because you described P<0.05 as significant, when in reality for 3 periods of analysis, this P value must be adjusted.

Results:

Patients were extremely different regarding demographic data and clinical data. Could this interfere with the results of comparing biomarkers in relation to identifying AKI?

In the first paragraph on page 13, you describe AUC results of the prognostic scores, I suggest removing them. They do not bring any news and are not related to the study subject.

Based on the analyzes performed, I am having difficulties in identifying the precocity of the biomarker in identifying AKI. Because we notice it elevated when the patient already has AKI and it rises more according to the severity of the case. Could you discuss this?

Discussion

Follow the model standardized by STROBE or another model to have a better organization of the discussion.

The final conclusion is better elucidated than the abstract one, especially on the issue of predicting AKI.

Reviewer #2: There are some minor points to improve.

In the description of the patients the presence of CKD without K-DIGO staging is reported in the table. Is it possible to insert in the demographic table the stages of the CKD? The timing of the sample is not homogeneous for all patients, as the first sample is done within 24h and the second the morning of the second day. The time interval between these two samples is not constant for all patients. Can you add a comment in the text?.

Is there a role of the CKD stage in urinary actin elimination? Can you provide a comment?

6. PLOS authors have the option to publish the peer review history of their article (what does this mean?). If published, this will include your full peer review and any attached files.

Reviewer #1: No

Reviewer #2: No

---

## [Author Response · Author response to Decision Letter 0]

13 Jul 2021

Response to reviewers

Urinary actin, as a potential marker of sepsis-related acute kidney injury: a pilot study

Dear Prof. Dr. Martina Crivellari,

First of all, we would like to say thank you for handling our manuscript as an academic editor of PLOS One! Based on the received valuable recommendations and comments, we made significant efforts to revise our manuscript. Please find our detailed answers for you and for the reviewers below.

Answers for Prof. Dr. Martina Crivellari:

• #1: In my personal opinion it is not clear how the patients were included.

Thank you for this comment! Please find the updated inclusion and exclusion criteria in the methods section of the revised manuscript.

The description of inclusion and exclusion criteria were slightly extended in the abstract as well (Row 30-38).

The sentence “Baseline Sequential Organ Failure Assessment (SOFA) scores were considered to be zero in each patient due to the lack of clinical data prior to ICU admission.” was deleted to clarify the inclusion criteria (Row 111-113).

We also corrected the inclusion criteria for sepsis: “Inclusion criteria for sepsis were signs of organ dysfunction shown in increased Sequential Organ Failure Assessment (SOFA) score (>2; median: 10), elevated serum PCT levels (>2 ng/mL; median: 9.865 ng/mL) and a suspected or microbiologically confirmed infection.” (Row 113-116)

• #2: There is no information about patient therapy.

Thank you for raising this important point, which will improve the quality of our manuscript. We added the following lines regarding therapy of patients to the methods and results sections of the revised manuscript:

“Management of sepsis and sepsis-related AKI followed the international guidelines of the 2016 Surviving Sepsis Campaign regarding vasopressor, respiratory, anticoagulation and hydrocortisone therapy, along with feeding, ulcer prophylaxis and sedation. All patients received adequate fluid resuscitation and ex juvantibus broad spectrum antimicrobial therapy guided by the clinical presentation for 24 to 72 hours, which was later modified based on the results of microbiological investigations.” (Row 120-124).

“Major differences in therapeutic requirements of 60 septic patients are the following: 38 needed mechanical ventilation 22 needed oxygen supplementation (median Horowitz quotient: 221.2), 51 needed vasopressor support, 12 had liver failure, 13 had thrombocytopenia, 30 had elevated lactate levels (>2 mmol/L) and 47 needed hydrocortisone supplementation. Ventilated patients received Propofol or Dexmedetomidine sedation during the early stage of severe respiratory failure” (Row 200-205) (S1 Table).

• #3: You should discuss the timing of this marker in identifying the AKI.

Thank you for this remark! Please find the following lines in the discussion section of the revised manuscript: “As actin is bound to gelsolin or Gc-globulin in the circulation, it is unlikely to appear in the urine. However, u-actin could appear in the urine due to both severe glomerular or tubular injury, so it seems that the elevation of u-actin assumes severe kidney injury. As KIM-1 or NGAL are known examples of damage biomarkers in AKI according to recent reviews, u-actin may also be classified into this group. We suspect that u-actin might provide more accurate information regarding the extent of kidney injury compared with serum creatinine. Therefore, a multi-marker approach including u-actin and the measurement of various damage biomarkers (e.g. urine Cystatin C, KIM-1, NGAL, IL-18) may provide valuable information regarding the more accurate staging of sepsis-related AKI.” (Row 342-350)

Dear Reviewer,

Thank you for your valuable time you spent on providing this precise and professional review regarding our manuscript. We are really grateful for all of your valuable recommendations and comments, and we made significant efforts to correct the manuscript accordingly! Please find our detailed answers below!

Reviewer #1 answers:

• #1: Title: why include a pilot study in the title? Nowhere in the text was that explained.

Thank you for this question! As far as we know, before our study nobody investigated u-actin in sepsis or in sepsis-related AKI, thus we aimed to be the first to explore this promising and interesting field. Since we were not sure on the connections and correlations between u-actin, sepsis and sepsis-related AKI, we decided to conduct a small sample size pilot study first to establish some baseline data in this unexplored field. These are the reasons why we consider our study as a pilot study. (Row 316-319).

• #2: Abstract: Instead of describing the sample size in the methods why didn't you describe which patients were included and excluded from the study? The sample size must be considered in the results.

Thank you for this remark! We rearranged the methods section of the abstract, which now includes more information regarding the inclusion and exclusion criteria of the patients (Row 30-37), while the sentence describing the sample size is now moved to the results section of the abstract (Row 38).

• #3: Introduction: The first paragraph separately presents the subject of sepsis and your manuscript is related to sepsis-AKI, so I suggest shortening the information in this paragraph, because it is out of context. In the introduction there is a lack of information about AKI biomarkers, it is subtly described.

Thank you for your suggestions! The introduction section is now corrected, namely the paragraph describing sepsis is shorter (Row 58-66), while there is more information about potential AKI biomarkers described in the literature (Row 76-82).

• #4: Methods: In this item we have the biggest weakness of the study. First of all, how were the patients included? consecutively? The study period is very long and the number of patients seems to be small, why did this happen? Was there a random choice of patients to participate in the study or as admitted were they involved in the study? This information is relevant because it can be a bias in patient selection.

Thank you for this very important comment.

The patients were included consecutively (Row 100)! However, we conducted our research as a single center study (16 bedded central ICU), thus we had very limited capacities for consecutive patient recruitment without randomization, hence the relatively long study period for this small number of patients (Row 319-322).

• #5: Inclusion and exclusion criteria are poorly described. Could patients with previous renal failure or dialysis participate in the study? Could patients in palliative care without treatment perspective also participate in the study? Could you better explain all these criteria and references for the decision on inclusion and exclusion? This issue impacts the final results found in the study.

Thank you for this remark! The inclusion and exclusion criteria were updated for septic patients in the revised manuscript. Inclusion criteria were shortened for easier understanding (Row 111-116). Exclusion criteria were updated in order to clarify that patients with malignancies needing palliative care, end-stage renal disease with chronic dialysis or kidney transplantation did not participate in the study (Row 104-106). Distinguishing the cause of kidney injury would have been harder without the exclusion of these conditions. 

• #6: Second point that is not clear, the sample size included in the study, why did this distribution occur: 17 sepsis, 43 sepsis-AKI and 24 controls? How were these numbers found? Are these numbers enough to achieve the results found?

Thank you for this comment! As the consecutive inclusion of eligible patients took a long time, we still could not achieve an equal distribution between the patient groups (Row 319-324). Since our central ICU is the regional center for the treatment of patients with severe sepsis, we could include more patients with sepsis-related AKI. The incidence of AKI is increasing significantly according to sepsis severity (4.2% for sepsis, 22.7% for severe sepsis, and 52.8% for septic shock) after Alobaidi R et al. (DOI: 10.1016/j.semnephrol.2015.01.002). However, if we divide the sepsis-related AKI group, then the AKI-1 (10 patients), AKI-2 (14 patients) and AKI-3 (19 patients) groups are more comparable to the 17 septic and 24 control patients. The 24 patients in the control group were without AKI and sepsis, therefore the number of this group was sufficient for statistical analysis.

• #7: Third, some relevant information needs to be collected in the study, for example fluid balance, use of vasopressor drugs and the need for mechanical ventilation. These issues could directly impact the outcomes, it is important to know if patients were treated equally and what the main protocols for these events would be.

Thank you for these comments! Please find the following lines regarding therapy of patients in the methods and results sections of the revised manuscript: 

“Management of sepsis and sepsis-related AKI followed the international guidelines of the 2016 Surviving Sepsis Campaign regarding vasopressor, respiratory, anticoagulation and hydrocortisone therapy, along with feeding, ulcer prophylaxis and sedation. All patients received adequate fluid resuscitation and ex juvantibus broad spectrum antimicrobial therapy guided by the clinical presentation for 24 to 72 hours, which was later modified based on the results of microbiological investigations” (Row 120-124).

“Major differences in therapeutic requirements of 60 septic patients are the following: 38 needed mechanical ventilation 22 needed oxygen supplementation (median Horowitz quotient: 221.2), 51 needed vasopressor support, 12 had liver failure, 13 had thrombocytopenia, 30 had elevated lactate levels (>2 mmol/L) and 47 needed hydrocortisone supplementation. Ventilated patients received Propofol or Dexmedetomidine sedation during the early stage of severe respiratory failure” (Row 200-205) (S1 Table).

• #8: Finally, why were these sample collection periods adopted for the realization of biomarkers? How were the correlations between biomarkers analyzed based on these different periods of material collection?

Thank you for this important comment! We hypothesized that u-actin concentrations would quickly elevate during an extensive tissue injury, e.g. during severe sepsis. To our best knowledge, there is no clear evidence regarding the kinetics of u-actin in sepsis (Row 316-319). In the future, we plan to extend the sampling period for 5-7 days (Row 367-369). The correlations between biomarkers were evaluated by Spearman's rank correlation test including all sample collection time points (T1-T3) (Row 275-276). These calculations were also performed regarding the admission laboratory data (T1) without any major differences.

(U-actin correlations for T1-T3: u-actin/u-creatinine (ρ=0.898, p<0.001); Moderate correlations (p<0.001): u-TP (ρ=0.489), u-albumin (ρ=0.617), se-creatinine (ρ=0.371), u-CysC (ρ=0.434) and u-ORM (ρ=0.367); Weak correlations (p<0.01): se-actin (ρ=0.272) and WBC (ρ=0.223))

(U-actin correlations for T1: u-actin/u-creatinine (ρ=0.872, p<0.001); Moderate correlations (p<0.001): u-TP (ρ=0.572), u-albumin (ρ=0.678), se-creatinine (ρ=0.575), u-CysC (ρ=0.491) and u-ORM (ρ=0.496); Weak correlations (p<0.01): se-actin (ρ=0.347) and WBC (ρ=0.344))

• #9: Statistical analysis: What was the outcome parameter used in ROC and how was the reference point of the markers reached? Could you describe this calculation better?

Thank you for this remark! Our scientific question was: Is u-actin a sensitive marker regarding the development of AKI in sepsis? The outcome parameter was the occurrence of AKI among the first day samples of septic patients, meaning that the admission data of all sepsis-related AKI patients (AKI-1, AKI-2, AKI-3) were compared with septic patients without AKI (Row 260-262). The difference in group sizes is a limitation regarding this calculation as well, yet the results from this non-parametric test showed a strong statistical significance (p<0.001) for u-actin and se-creatinine.

• #10: In the analysis of multiple comparisons, the value of P was not corrected? Because you described P<0.05 as significant, when in reality for 3 periods of analysis, this P value must be adjusted.

Thank you for this comment! Indeed, the P values were adjusted according to Bonferroni during the analysis of multiple comparisons, hence achieving the results shown during the data comparison of different time points. (Row 180-181)

• #11: Results: Patients were extremely different regarding demographic data and clinical data. Could this interfere with the results of comparing biomarkers in relation to identifying AKI?

Thank you for this remark! As we already mentioned, we had limited capacities in many ways regarding this pilot study, hence we could investigate only a relatively small number of patients (Row 319-324). This is a limitation of our study, yet we assume that these differences do not interfere with the correct interpretation of our results, e.g. we did not see any significant correlation between u-actin and creatine kinase or myoglobin which could hint that differences in muscle mass due to age or gender would not influence the results. According to the most recent Sepsis-3 definitions, the incidence of sepsis is significantly increased in newborns, older patients and males (DOI:10.1001/jama.2016.0287). Type-2 diabetes mellitus or immunological disorders may influence the incidence of renal failure, yet there were no significant differences in our patient groups regarding these comorbidities.

• #12: Based on the analyzes performed, I am having difficulties in identifying the precocity of the biomarker in identifying AKI. Because we notice it elevated when the patient already has AKI and it rises more according to the severity of the case. Could you discuss this?

Thank you for this comment! We were hypothesizing the possible sources of u-actin elevation. “As actin is bound to gelsolin or Gc-globulin in the circulation, it is unlikely to appear in the urine. However, u-actin could appear in the urine due to both severe glomerular or tubular injury, so it seems that the elevation of u-actin assumes severe kidney injury. As KIM-1 or NGAL are known examples of damage biomarkers in AKI according to recent reviews, u-actin may also be classified into this group. We suspect that u-actin might provide more accurate information regarding the extent of kidney injury compared to serum creatinine. Therefore, a multi-marker approach including u-actin and the measurement of various damage biomarkers (e.g. urine Cystatin C, KIM-1, NGAL, IL-18) may provide valuable information regarding the more accurate staging of sepsis-related AKI” (Row 342-350).

• #13: Discussion: Follow the model standardized by STROBE or another model to have a better organization of the discussion.

Thank you for your advice, the discussion section is reorganized according to the STROBE model for easier understanding (e.g. the limitations section was moved below the summary of our results (Row 316-326)).

• #14: The final conclusion is better elucidated than the abstract one, especially on the issue of predicting AKI.

Thank you for your remark, the content of the conclusion section is partially shown in the conclusion section of the abstract (Row 48-50).

Dear Reviewer,

Thank you for your time and for all the valuable comments and recommendations! We made significant efforts to revise our manuscript accordingly. Please find our detailed answer below!

Reviewer #2 answers:

• #1: In the description of the patients the presence of CKD without K-DIGO staging is reported in the table. Is it possible to insert in the demographic table the stages of the CKD?

Thank you for this recommendation! The necessary parameters (KDIGO staging of patients with CKD) are now added to the demographic table (Table 1) while the table provided as supplemental material (S1 Table) was corrected as well.

• #2: The timing of the sample is not homogeneous for all patients, as the first sample is done within 24h and the second the morning of the second day. The time interval between these two samples is not constant for all patients. Can you add a comment in the text?

Thank you for this remark! The majority of first day samples were collected on the next morning after admission, mostly within 12 hours, since most patients were admitted either at night or in the late afternoon before sample collection. This means that there was a slightly variable time interval before taking the first sample, which is a limitation of our study, yet we assume that these differences do not interfere with the correct interpretation of our results (Row 324-326).

• #3: Is there a role of the CKD stage in urinary actin elimination? Can you provide a comment?

Thank you for this question! Patients with end-stage renal disease were excluded from our study, yet we hypothesize that actin could appear in the urine due to severe glomerular injury. In order to clarify this issue, we should involve more non-septic CKD patients in the future (Row 370-372).

---

## [Editor Report · Decision Letter 1]

14 Jul 2021

Urinary actin, as a potential marker of sepsis-related acute kidney injury: a pilot study

PONE-D-21-17545R1

Dear Dr. Ragàn,

We’re pleased to inform you that your manuscript has been judged scientifically suitable for publication and will be formally accepted for publication once it meets all outstanding technical requirements.

Kind regards,

Martina Crivellari

Academic Editor

PLOS ONE
---

## [Editor Report · Acceptance letter]

16 Jul 2021

PONE-D-21-17545R1 

Urinary actin, as a potential marker of sepsis-related acute kidney injury: a pilot study 

Dear Dr. Ragán:

I'm pleased to inform you that your manuscript has been deemed suitable for publication in PLOS ONE. Congratulations! Your manuscript is now with our production department. 

Kind regards, 

on behalf of

Dr. Martina Crivellari 

Academic Editor

PLOS ONE